# QDC: Quantum Diffusion Convolution Kernels on Graphs

**Thomas Markovich**  *tmarkovich@squareup.com*
*CashApp*
*Cambridge, Massachusetts, USA*

**Reviewed on OpenReview:** *https://openreview.net/forum?id=uXGUSX8GoY*

## Abstract

Graph convolutional neural networks (GCNs) operate by aggregating messages over local neighborhoods given the prediction task under interest. Many GCNs can be understood as a form of generalized diffusion of input features on the graph, and significant work has been dedicated to improving predictive accuracy by altering the ways of message passing. In this work, we propose a new convolution kernel that effectively rewires the graph according to the occupation correlations of the vertices by trading on the generalized diffusion paradigm for the propagation of a quantum particle over the graph. We term this new convolution kernel the Quantum Diffusion Convolution (QDC) operator. In addition, we introduce a multiscale variant that combines messages from the QDC operator and the traditional combinatorial Laplacian. To understand our method, we explore the spectral dependence of homophily and the importance of quantum dynamics in the construction of a bandpass filter. Through these studies, as well as experiments on a range of datasets, we observe that QDC improves predictive performance on the widely used benchmark datasets when compared to similar methods.

## 1 Introduction

Graphs are mathematical structures composed of vertices and edges, where the edges represent potentially complex relationships or interactions between the vertices. Graph structured data involves complex patterns and relationships that can not be captured by the traditional deep learning methods that focus on tabular data. As a result, graph structured data and graph machine learning models have become increasingly important in many fields, such as machine learning (Wu et al., 2020), computer vision (Krzywda et al., 2022), and natural language processing (Wu et al., 2023). Indeed, graphs and graph structured data are ubiquitous in industrial applications ranging from fraud detection (Liu et al., 2020; Zhang et al., 2022), to routing (Rusek et al., 2019; Chen et al., 2022), weather predictions (Keisler, 2022; Ma et al., 2022), drug discovery (Bongini et al., 2021; Han et al., 2021; Xiong et al., 2021), and personalized recommendations (Wu et al., 2022; Gao et al., 2021).

Graph neural networks (GNNs) are an increasingly popular modality for constructing graph machine learning models (Zhou et al., 2020; Wu et al., 2020). There are many variations of architectures for GNNs, but most GNNs can be thought of as having a function that, for a given vertex, aggregates information from its neighbors, and a second function which maps this aggregated information to the machine learning task under investigation, such as node classification, node regression, or link prediction. A simple but powerful model is a Graph Convolutional network (GCN), which extends the convolutional neural network (CNN) architecture to the graph domain by using a localized filter that aggregates information from neighboring nodes (Zhang et al., 2019). By sharing weights across different nodes, GCNs can learn representations that capture both the local and global structure of the graph. These models have shown remarkable success in a variety of tasks such as node classification (Kipf & Welling, 2016; Zhang et al., 2019), graph classification(Xie et al., 2020), community detection (Jin et al., 2019; Wang et al., 2021), and link prediction (Chen et al., 2020; Cai et al., 2019; Zeb et al., 2022). GCNs learn filters on the graph structure to be used at inference time. Early GCN development learned these filters in the spectral domain (Bruna et al., 2013), but this requires

the decomposition of large matrices. Due to the computational expense of these decompositions, spatial filters rose in popularity and have been the dominant paradigm. Significant effort has been dedicated to methodological improvements that make spatial convolutions more expressive (Bodnar et al., 2021; Bouritsas et al., 2022) and scalable (Hamilton et al., 2017; Ying et al., 2018).

Further work has shown that it is possible to unify many of these models based on spatial convolutions as generalized graph diffusion (Chamberlain et al., 2021a;b; Gasteiger et al., 2019), with considerable attention being focused on improving diffusion dynamics (Elhag et al., 2022; Di Giovanni et al., 2022). These generalized graph diffusion models can be understood variants of the heat diffusion equation. Heat diffusion on graphs is well known to thermalize quickly, leading all vertices to average to the same temperature value. This physical phenomenon is similar to the oversmoothing behaviour that has plagued GCNs. There have been many approaches to combat this oversmoothing by altering the properties of the diffusion while keeping the underlying diffusion dynamics (Alon & Yahav, 2020; Zhu et al., 2020; Topping et al., 2021).

This work is inspired by Gasteiger et al. (2019) and the recognition that the Graph Diffusion Convolution method (GDC) rewires the graph using a low-pass filter that is inspired by heat diffusion. We build upon this foundation by considering the question "Can we improve graph neural networks by rewiring the base graph with a learnable band-pass filter?". We find our answer in a rewiring method that is structured around a learnable band-pass filter and inspired by the Schrödinger equation. Following in the steps of Gasteiger et al. (2019), we construct a graph Laplacian preprocessing framework that captures the band-limited infinite-time dynamics of quantum diffusion through a system. We call this framework QDC, and find that this framework is very flexible and can be included into the architecture of many graph neural networks. In summary, this paper's core contributions are

- We propose QDC, a quantum mechanically inspired diffusion kernel, a more powerful and general method for computing sparsified non-local transition matrices.

- We propose a novel multi-scale message passing paradigm that performs message passing using QDC and the original combinatorial Laplacian in parallel.

- We compare and evaluate QDC and MultiScaleQDC to a set of similar baselines on a range of node classification tasks.

- We analyze the spectral dependence of homophily in graph datasets and show that many heterophilous datasets are actually homophilous in filtered settings.

## 2 Related Work

Our method can be viewed as a technique for graph rewiring, because it changes the computation graph from the original adjacency matrix to a filtered one. Graph rewiring is a common technique in the literature for improving GNN performance by removing spurious connections. The Stochastic Discrete Ricci Flow (SDRF) method seeks to perform this rewiring by fixing instances of negative curvature on the graph (Topping et al., 2021). The Fully Adjacent layer method (+FA) attempts to improve performance by adding a fully adjacent layer to message passing (Alon & Yahav, 2020). GDC is the most similar method to ours, but is based on the idea of heat, rather than quantum, diffusion which yields a low pass filter (Gasteiger et al., 2019).

Our method can also be understood within the context of physics inspired graph neural networks. Graph-HEAT proposes performing graph convolutions using a parameterized version of the heat kernel (Xu et al., 2020). The Graph Neural Diffusion (GRAND) method recasts message passing as anisotropic diffusion on a graph, and provides a framework with which to unify many popular GNN architectures (Chamberlain et al., 2021a; Thorpe et al., 2022). BLEND pushes this perspective further to explore diffusion in non-euclidean domains (Chamberlain et al., 2021b). PDE-GCN looks further and seeks to combine diffusion with the wave equation to define new message passing frameworks (Eliasof et al., 2021). To our knowledge, ours is the first work that explores quantum dynamics as a message passing formalism.

Our method is closely related to the kernel signatures that have been explored in computer graphics. The Heat Kernel Signature was one of the first kernel signatures that was developed, and was developed by

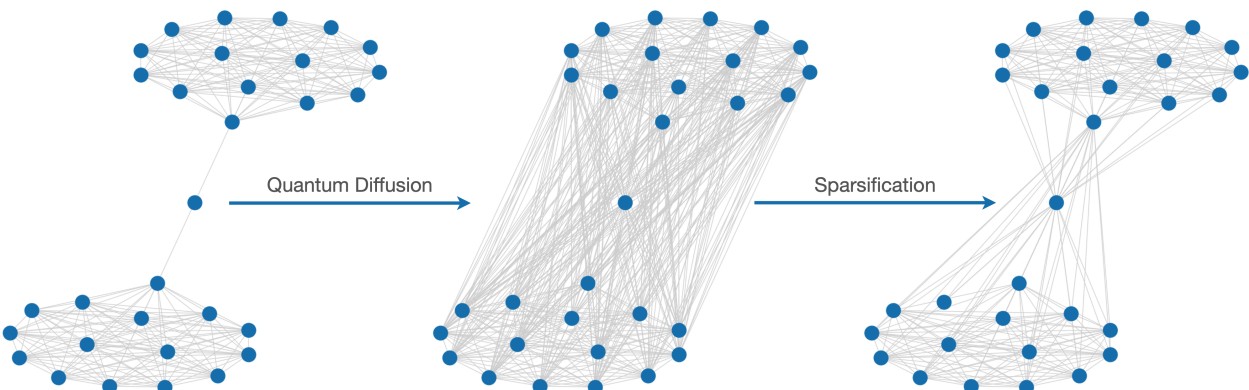

Figure 1: An illustration of the Quantum Diffusion Convolution Kernel. The graph is transformed through the propagation of a quantum particle as given by equation **??**, and then sparsified the graph to account for the presence of rare events. The model is then trained on the post-sparsified rewired graph.

modeling the heat diffusion over a surface (Sun et al., 2009; Zobel et al., 2011). It was observed that the Heat Kernel Signature was sensitive to motions that might introduce a bottle neck, such as an articulation, which motivated the development of the Wave Kernel signature. The Wave Kernel Signature defines surface structural features by modeling the infinite time diffusion of quantum particles over the surface (Aubry et al., 2011). Building on the wave kernel signature, the average mixing kernel signature explores the finite time diffusion of a quantum particle over the surface (Cosmo et al., 2020). The wave kernel trace method instead explores solutions of the acoustic wave equation using the edge-Laplacian (Aziz et al., 2014). These methods have all been used to develop graph features that would then be used to identify similar shapes, but we are instead using these kernels as convolution operators.

## 3 Background

**Preliminaries** Given an undirected graph, $\mathcal{G} = (\mathcal{V}, \mathcal{E}, \boldsymbol{X})$, where $\mathcal{V}$ is the vertex set with cardinality $|\mathcal{V}| = N$, and $\mathcal{E}$ is the edge set, and $\boldsymbol{X} \in \mathbb{R}^{Nxd}$ denote the matrix of vertex features, where d is the dimensionality of the feature set. $\mathcal{E}$ admits an adjacency matrix, $\boldsymbol{A} \in \mathbb{R}^{NxN}$, where $A_{ij} = 1$ if and only if vertices $i$ and $j$ are connected. Because we have restricted ourselves to undirected graphs, $A_{ij} = A_{ji}$. While this $\mathcal{G}$ could have weighted edges, we focus on the unweighted case for simplicity. It is common to augment a graph with self loops, which is performed by $\tilde{\boldsymbol{A}} = \boldsymbol{I} - \boldsymbol{A}$, to allow for message passing of depth $l$ to include messages from all random walks of length $r \leq l + 1$. With these definitions in hand, we define our combinatorial graph Laplacian as $\mathcal{L} = \boldsymbol{D}^{-\frac{1}{2}} \tilde{\boldsymbol{A}} \boldsymbol{D}^{-\frac{1}{2}}$, where $\boldsymbol{D}^{-\frac{1}{2}}$ is the diagonal degree matrix of $\tilde{\boldsymbol{A}}$. Other normalizations are possible, but we restrict our discussion to symmetrically normalized graph Laplacians.

**Graph Signal Processing** The central challenge of signal processing on graphs is to apply the intuition from signal processing on structured grids to nonuniform and unstructured graph domains. We do so by drawing an analogy between graph and rectangular domains. The Fourier basis in rectangular domains is given by $\phi(\boldsymbol{x}, \boldsymbol{k}) = e^{i\boldsymbol{k} \cdot \boldsymbol{x}}$, where $\boldsymbol{k}$ is the vector of frequencies, or wave numbers, and $\boldsymbol{x}$ is the vector of positions, which are the eigenfunctions of the Laplacian in rectangular coordinates. The definition of the Fourier basis allows us to define the Fourier transform and the convolution theorem:

$$f(x) = \{g * h\} = \mathcal{F}^{-1} \left\{ \mathcal{F}(g) \cdot \mathcal{F}(h) \right\}, \tag{1}$$

where $\mathcal{F}$ and $\mathcal{F}^{-1}$ is the Fourier transform and inverse Fourier transform respectively; $*$ is the spatial convolution; and $\cdot$ is the point wise product. Convolutions are ubiquitous in signal processing, with spectral filtering being the application that we will focus on. From the perspective of spectral filtering, we have a signal, $h(x)$, and a filter given by $g(\cdot)$. We can either apply this filter in the spectral or conjugate spatial

domain, but spectral filtering is preferred. If the optimal filter is unknown *a priori*, we can learn this filter through a learning procedure. This can be done in a variety of ways, such as learning the filter in the Fourier domain. This amounts to learning a filter with infinite support and an infinite number of free parameters. While theoretically possible, this is a suboptimal technique. Instead, we frequently choose to expand into a convenient basis. In this way, we end up with:

$$f(x) = \left\{ \sum_k (c_k g_k) * h \right\} = \mathcal{F}^{-1} \left\{ \sum_k c_k \mathcal{F}(g_k) \cdot \mathcal{F}(h) \right\}, \tag{2}$$

where $c_k$ is the expansion coefficient for each basis function given by $g_k$. Choices of basis function can include gaussians, wavelets, polynomials, or other functional forms depending on the desired properties. The implicit assumption with this filtering construction is that there are components of the signal which are spurious for a given application. Depending on the application and the signal, we might need to learn a filter that damps high or low frequency noise, or only selects a narrow set of frequencies. These are called low-pass, high-pass, and band-pass filters respectively.

Analogously, we can use the definition of the graph Laplacian to construct a Fourier basis in the graph domain. In the case of an undirected graph, the Laplacian is a symmetric positive semidefinite matrix which admits an eigensystem with orthonormal eigenvectors and real, positive, eigenvalues. These eigenvectors form the graph Fourier basis and the eigenvectors form the squared frequencies. We can then write the eigendecomposition as $\mathcal{L} = \boldsymbol{U}^T \Lambda \boldsymbol{U}$, where $\boldsymbol{U}$ is the matrix of eigenvectors and $\Lambda$ is the diagonal matrix of eigenvalues.

This definition allows us to begin to define filters. The original spectral GCN paper Bruna et al. (2013) learned filters of the form $g(\Theta) = \text{diag}(\Theta)$, where $\text{diag}(\theta)$ is learned set of parameters for each of the Fourier basis functions. This is then used for filtering as:

$$f(x) = U \left[ g(\Theta) \cdot \left( U^T x \right) \right] \tag{3}$$

While effective and highly flexible, this filtering technique has $O(n)$ learnable parameters, can be difficult to learn, and requires decomposition of $\mathcal{L}$ to construct eigenvectors $\boldsymbol{U}$. Spectral convolutions are mathematically elegant, and the construction of the basis and learning of the filter are computationally demanding, making them not ideal in many use cases. By contrast, convolutions can also be defined in the spatial domain. Spatial convolutions are spatially localized, so they have finite size, but they aren't guaranteed to be unique which makes them computationally difficult to learn and to apply. This method makes the assumption that we can approximate $g$ as $g(\Lambda; \{\Theta\}) = \sum_k^K \theta_k \Lambda^k$, which yields a filtered signal of the form:

$$f(x) = \sum_k^K \theta_k \mathcal{L}^k x. \tag{4}$$

This learned filter is more parameter efficient and spatially localized, because convolutions only extend to a distance reachable by $K$ applications of the Laplacian. Given that we desire to limit our computational cost, we will restrict ourselves to learnable functional forms that result only in matrix products because for sparse matrices, these scale as $O(|\mathcal{E}|)$. Chebyshev polynomials are one straightforward choice because they are the interpolating polynomial that provides the best function approximation under the maximum error norm. These polynomials yield a filter of the form $g(\Lambda; \{\Theta\}) = \sum_k^K \theta_k T_k(\tilde{\Lambda})$, where $T_k$ is the $k^{th}$ Chebyshev polynomial given by the recurrence relationship $T_k(x) = 2x T_{k-1}(x) - T_{n-2}(x)$, and $\tilde{\Lambda}$ is the rescaled eigenvalue matrix (Defferrard et al., 2016).

**Diffusion as Convolution** We can readily identify equation 4 as the Taylor expansions of the matrix exponential for particular choices of $\theta$, which has a similar form to the Heat kernel. This amounts to solving the partial differential equation;

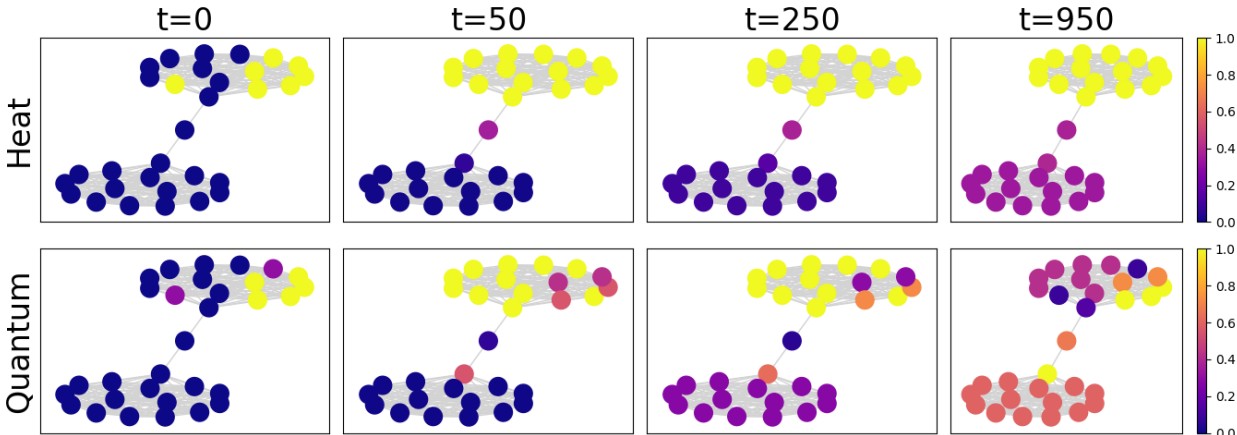

Figure 2: A comparison of both heat diffusion and quantum dynamics on a barbell graph. The top row corresponds diffusion according to the heat equation of the graph, and the bottom row corresponds to propagation of the Schrödinger equation. We simulated dynamics for 1000 unitless timesteps with the same initial distribution for both equations. We use the `plasma` colorbar, with blue and yellow corresponding to the minimum and maximum respectively. We observe the top row thermalizes within the cluster rapidly to the same temperature but encounters a bottleneck as the two ends of the barbell thermalize. By contrast, quantum dynamics exhibits oscillatory behavior both within the clusters as well as across the clusters, and probability density migrates rapidly.

$$\frac{\partial f}{\partial t} = -c \triangle f = -c\mathcal{L}f, \tag{5}$$

where $\triangle$ is the Laplace-Beltrami on the graph, and admits a solution $f(t) = \exp(-t\mathcal{L})f(0)$. Viewed in this way, we can draw the equivalence between solving the heat equation on the graph and many message passing schemes. Building on this observation, Gasteiger et al. (2019) observed that it is possible to define a generalized graph diffusion operator $\boldsymbol{S}$ as $\boldsymbol{S} = \sum_k^\infty \theta_k \boldsymbol{T}^k$ where $\boldsymbol{T}$ is a transition matrix, such as the combinatorial Laplacian. As we can see from the definition of the heat kernel, it exponentially squashes the contributions from eigenmodes with large eigenvalues, thereby providing us with a low-pass filter. Intuitively, this should help in settings where averaging out the noise from neighbors can improve performance. Following this line of thought, many graph neural network architectures can be understood as performing generalized diffusion over a graph (Chamberlain et al., 2021b;a).

From a physical perspective, the heat equation is well known to drive towards an infinitely smooth equilibrium solution; by smoothing out the initial distribution exponentially quickly in acausal ways. The heat equation is also well known to exhibit thermal bottlenecks when the heat flux and temperature gradient vectors deviate (Bornoff et al., 2011; Grossmann & Lohse, 2000). This physical process is analogous to the oversmoothing and oversquashing problems that have plagued graph neural networks respectively. We can observe both of these physical processes in Figure 2, which presents a comparison of heat and quantum diffusion at four different time steps. By $t = 50$, we observe that the top lobe of the barbell has completely thermalized, or oversmoothed, and heat is slowly starting to leak into the center vertex, or oversquashed. In 200 timesteps, we observe very little in the way of heat transfer. It is clear that there is a thermal bottle neck in our system, and this is hampering the flow of heat.

## 4 Methodology

**Quantum Convolution Kernels** Because the graph defines both the geometry of the system and its Laplacian, it is natural to ask if grounding our rewiring in a different physical model would yield better results. There are many possible options including the wave equation, which propagates the sharpness of

solutions; the elastic wave equation which models both forces as displacements as vector quantities; the convection equation which models the flux; the eikonal equation which simulates shortest path searches; or Schrödinger's equation which models quantum particles. Of these partial differential equations (PDEs), we turn our attention to Schrödinger's equation. While structurally similar to the heat equation, its dynamics are stable, do not lead to oversmoothing, natively capture both constructive and destructive interference, and are controllable through the engineering of potential energy surfaces (Chou et al., 2012; Jaffé, 1987; Sakurai & Commins, 1995). Indeed, Schrödinger's equation and the heat equation are related through a Wick rotation by $\pi/2$ (Popov, 2005). Qualitatively, we observe in Figure 2 that unlike thermal diffusion, quantum diffusion is able to quickly pass information through the bottleneck. Later timesteps show us constructive and destructive interference that provide structure to the propagations. Videos of the propagation show oscillations across the top lobe and along the vertical axis as a function of time. Qualitatively, these dynamics do not seem to oversmooth and are less prone to oversquashing as a result. Motivated by these qualitative results, we briefly introduce Schrödinger's equation, which are used to derive our quantum convolution kernel. The dynamics are governed by the time dependent Schrödinger equation, which, for a free particle, is given by;

$$i\frac{\partial \psi(x,t)}{\partial t} = -\triangle \psi(x,t). \tag{6}$$

On a graph, the eigenstates of $\triangle$ define a finite, complete, and orthogonal basis where the position arguments correspond to the vertex locations. Therefore, the expectation value $\langle \psi(x_i, 0), \psi(x_j, t) \rangle$ computes the probability that a particle on vertex $i$ is found on vertex $j$ at some time $t$. In our setting, we wish to compute the long-term steady-state distribution, so we compute the energy-filtered infinite time integral of the average overlap between any two vertices, which is given by;

$$\int_0^\infty dt \frac{1}{\sigma} e^{-\frac{\sigma^2 t^2}{2} + i\mu t} \psi(x_i, 0)^\dagger \psi(x_j, t) = \sum_\alpha e^{-(E_i - \mu)^2/2\sigma^2} \phi_\alpha^\dagger(x_i)\phi_\beta(x_j) = \mathcal{Q}(x_i, x_j), \tag{7}$$

where $\mathcal{Q}$ is our Quantum Diffusion Kernel (QDC), and both $\mu$ and $\sigma$ are our two tunable parameters. Because GCNs learn spectral filters, we observe that this rewiring process is frequency dependent and has two learnable parameters in the diffusion kernel. Intuitively, we interpret $Q(x_i, x_j)$ as the time averaged probability of transition from vertex $i$ to vertex $j$. Analogously to GDC, we can use $\mathcal{Q}$ as our transition matrix, instead of combinatorial graph Laplacian. Doing so allows us to use QDC with any message passing neural network by simply replacing $\mathcal{L}$ with $\mathcal{Q}$. We present a schematic of our method in Figure 1 in which the first transformation corresponds to switching from the original graph laplacian to transition matrix.

**Sparsification** QDC defined as $\mathcal{Q}(x_i, x_j)$ is a matrix $\mathcal{Q}_{i,j} = \mathcal{Q}(x_i, x_j)$, where $\mathcal{Q}_{i,j}$ is the probability of transition from vertex $i$ to vertex $j$. Most graph diffusion results in a dense transition matrix, and QDC is no different. This happens because a quantum particle starting at site $i$ will visit all vertices within its connected component given an infinite amount of time, yielding probabilities that can be small but non-zero. This is a potentially major downfall of QDC when compared against spatial methods like Graph Diffusion Convolution (Gasteiger et al., 2019). This has the potential to introduce $\mathcal{O}(N^2)$ storage costs. To address this issue, we sparsify the resulting QDC matrix. We consider two different sparsification methods: a simple threshold based approach, or an approach that only keeps the top-k highest weighted connections. We denote the sparsified QDC kernel as $\tilde{\mathcal{Q}}$. While $\mathcal{Q}$ was both row and column normalized, $\tilde{\mathcal{Q}}$ is not. Therefore, after sparsification we normalize $\tilde{Q}$ in the usual way, defining $\tilde{Q}_{sym} = \boldsymbol{D}_{\tilde{Q}}^{-1/2}\tilde{Q}\boldsymbol{D}_{\tilde{Q}}^{-1/2}$. We will drop the *sym* in the following, such that all uses of $\tilde{Q}$ are normalized.

**Efficient Diagonalization** QDC is a spectral method, and depends on the eigendecomposition of $\mathcal{L}$. This is commonly viewed as too computationally demanding of a procedure because the full eigendecomposition of a matrix requires $\mathcal{O}(N^3)$ time, and the storage costs of the resulting dense eigensystem are $\mathcal{O}(N^2)$ where $N$ is the number of vertices. While this is generally true, we recognize from the form of our kernel in equation **??**, we are constructing a band pass filter and are thus only interested in a subset of the eigensystem. As a result, we are able to use approximate methods that are more computationally efficient.

Due to the importance of eigendecomposition to the computational sciences, this problem has received considerable attention with algorithms such as power iteration (Mises & Pollaczek-Geiringer, 1929), divide and conquer (Cuppen, 1980), Arnoldi iteration (Arnoldi, 1951), Lanczos iteration (Lanczos, 1950; Ojalvo & Newman, 1970), and LOBPCG Knyazev (2001); Knyazev et al. (2007). In this work we used LOBPCG, or Locally Optimal Block Preconditioned Conjugate Gradient, because it is provides a straightforward method to compute a limited number of eigenvectors and eigenvalues while only depending on the computation of matrix vector-products. In our applications, we use the folded spectrum method (MacDonald, 1934) along with LOBPCG to compute eigenvalues centered around $\mu$. If the solver is unable to converge, we retry with $\mu' = \mu + \epsilon_\lambda$, where $\epsilon_\lambda = 1e - 6$. In our settings, we compute $\min(512, N)$ eigenvalue, eigenvector pairs. In Table 2, we present average runtimes for training and testing for all GCN based methods. While we observe a significant increase in runtime, we attribute the majority of that cost to the preprocessing of the Laplacian, which requires both an eigendecomposition and sparse matrix multiply. In applied settings, it may be possible to cache the Laplacian which would allow the amortizaiton of this cost. These techniques do not change the fact that diagonalization, even with matrix free methods, is expensive.

**Multiscale GNN** QDC can be used as a drop-in replacement for a transition matrix for any message passing GNN. In section 5, we explore using QDC in place of $\mathcal{L}$ for both graph convolutional networks and graph attention networks. Because QDC provides a band pass filter, unlike GDC which provides a low-pass filter, it is interesting to explore the message passing across both $\mathcal{L}$ and $\mathcal{Q}$ in parallel. In this setting, we pass messages in parallel using $\mathcal{L}$ on one side and $\mathcal{Q}$ on the other. We then combined messages from each tower by either adding or concatenating them together. Finally, we feed the resulting messages into a readout function for predictions. We term this method MultiScaleQDC, because we are able to pass messages across multiple length scales of the graph.

## 5 Experiments

We tested our method on the task of semi-supervised node classification in both homophilic and heterophilic settings to understand the performance of our method. Broadly, we observed that QDC and MultiScaleQDC provided significant performance improvements upon the baseline methods that they were incorporated into. In some cases, the improvements can be as large as 20%. To understand these results, we analyze the spectral dependence of homophily and observe that QDC and MultiScaleQDC are able to rewire the graph in a way that makes it more homophilic. In the remainder of this section we present the details for these experiments as well as a more detailed discussion of the results.

**Experiment Setup** In an effort to ensure a fair comparison, we optimized the hyper-parameters of all models presented in Table 1 on all data sets. We performed 250 steps of hyper-parameter optimization for each method, including baselines, and the hyper-parameter search was performed using OPTUNA, a popular hyper-parameter optimization framework. All tuning was performed on the validation set, and we report the test-results associated with the hyper-parameter settings that maximize the validation accuracy. The parameters, and the distributions from which they were drawn, are reported in Appendix A.4. All experiments were run using PYTORCH GEOMETRIC 2.3.1 and PYTORCH 1.13, and all computations were run on an `Nvidia DGX A100` machine with 128 `AMD Rome 7742` cores and 8 `Nvidia A100` GPUs.

Because our method can be viewed as a Laplacian preprocessing technique, we use QDC in place of the traditional Laplacian in both graph convolution networks (GCN) (Kipf & Welling, 2016; Zhang et al., 2019), graph attention networks (GAT) (Veličković et al., 2018), and H₂GCN, a GNN designed to specifically handle heterophilic datasets (Zhu et al., 2020). QDC is similar in structure to graph diffusion convolution (GDC) (Gasteiger et al., 2019) and SDRF (Topping et al., 2021), so we have chosen to compare QDC to both GDC and SDRF in addition to an unprocessed Laplacian in a GCN, a GAT, and a H₂GCN. Our GCN and GAT models are implemented using the relevant layers from PYTORCH GEOMETRIC, and H₂GCN was run using an open source reimplementation (GitEventhandler, 2022). Similarly, our GDC implementation uses the publicly available version in PYTORCH GEOMETRIC, and we used the reference SDRF implementation from the authors (Topping, 2022).

**Datasets** We evaluated our method on 9 data sets: CORNELL, TEXAS, and WISCONSIN from the WebKB dataset; CHAMELEON and SQUIRREL from the Wiki dataset; ACTOR from the film dataset; and citation graphs

Table 1: Dataset statistics and experimental results on common node classification benchmarks. $\mathcal{H}$, $|\mathcal{V}|$, $|\mathcal{E}|$ denote degree of homophily, number of vertices and number of edges, respectively. Top results for each of the GCN and GAT families are highlighted in bold.

| | Cornell | Texas | Wisconsin | Chameleon | Squirrel | Actor | Cora | Citeseer | Pubmed |
|---|---|---|---|---|---|---|---|---|---|
| $\mathcal{H}$ | 0.11 | 0.06 | 0.16 | 0.25 | 0.22 | 0.24 | 0.83 | 0.71 | 0.79 |
| $|\mathcal{V}|$ | 183 | 183 | 251 | 2,277 | 5,201 | 7,600 | 2,708 | 3,327 | 18,717 |
| $|\mathcal{E}|$ | 280 | 295 | 466 | 31,421 | 198,493 | 26,752 | 5,278 | 4,676 | 44,327 |
| GCN | $45.68 \pm 7.30$ | $63.51 \pm 5.70$ | $59.22 \pm 4.28$ | $41.16 \pm 1.71$ | $27.89 \pm 1.21$ | $29.32 \pm 1.07$ | $87.46 \pm 1.11$ | $76.61 \pm 1.28$ | $\mathbf{88.47 \pm 0.39}$ |
| GCN+GDC | $47.03 \pm 5.69$ | $63.51 \pm 6.07$ | $57.25 \pm 2.88$ | $40.42 \pm 2.93$ | $27.97 \pm 0.93$ | $29.14 \pm 0.91$ | $87.63 \pm 0.91$ | $76.58 \pm 1.21$ | $88.46 \pm 0.55$ |
| GCN+SDRF | $45.14 \pm 8.20$ | $62.97 \pm 5.55$ | $57.84 \pm 1.52$ | $40.55 \pm 1.52$ | $28.17 \pm 0.97$ | $29.07 \pm 1.03$ | $87.44 \pm 1.10$ | $\mathbf{76.85 \pm 1.47}$ | $88.47 \pm 0.34$ |
| GCN+BPDC | $60.81 \pm 5.95$ | $68.92 \pm 6.54$ | $63.73 \pm 5.28$ | $50.44 \pm 1.77$ | $40.37 \pm 1.17$ | $31.46 \pm 1.04$ | $85.86 \pm 1.17$ | $74.70 \pm 1.34$ | $84.55 \pm 0.56$ |
| GCN+QDC | $63.78 \pm 9.68$ | $72.70 \pm 6.67$ | $65.29 \pm 6.80$ | $\mathbf{53.22 \pm 1.56}$ | $40.62 \pm 1.94$ | $\mathbf{35.08 \pm 0.64}$ | $86.00 \pm 1.56$ | $75.10 \pm 1.52$ | $84.65 \pm 0.44$ |
| GCN+MultiScaleQDC | $\mathbf{66.22 \pm 5.44}$ | $\mathbf{73.78 \pm 4.53}$ | $64.71 \pm 4.47$ | $54.71 \pm 2.79$ | $\mathbf{42.24 \pm 1.73}$ | $30.55 \pm 1.45$ | $\mathbf{87.85 \pm 0.85}$ | $76.72 \pm 1.49$ | $88.32 \pm 0.47$ |
| GAT | $60.81 \pm 8.40$ | $68.11 \pm 5.24$ | $63.14 \pm 7.58$ | $44.89 \pm 1.64$ | $31.47 \pm 1.44$ | $30.48 \pm 1.17$ | $86.68 \pm 1.64$ | $75.64 \pm 1.55$ | $84.11 \pm 0.70$ |
| GAT+GDC | $61.89 \pm 7.30$ | $68.11 \pm 5.09$ | $63.33 \pm 3.62$ | $45.96 \pm 1.94$ | $31.66 \pm 1.21$ | $31.18 \pm 0.76$ | $86.46 \pm 1.20$ | $75.92 \pm 1.10$ | $87.53 \pm 0.55$ |
| GAT+SDRF | $59.19 \pm 6.33$ | $67.30 \pm 4.90$ | $63.92 \pm 5.20$ | $45.88 \pm 1.93$ | $31.76 \pm 1.00$ | $31.13 \pm 0.76$ | $85.29 \pm 1.34$ | $75.90 \pm 1.27$ | $87.47 \pm 0.48$ |
| GAT+QDC | $\mathbf{77.57 \pm 3.83}$ | $\mathbf{87.57 \pm 5.56}$ | $\mathbf{88.04 \pm 3.33}$ | $50.90 \pm 2.16$ | $35.38 \pm 1.81$ | $35.57 \pm 1.05$ | $84.68 \pm 1.54$ | $75.21 \pm 1.30$ | $87.55 \pm 0.31$ |
| GAT+MultiScaleQDC | $77.03 \pm 4.05$ | $86.22 \pm 5.60$ | $88.04 \pm 4.06$ | $\mathbf{52.08 \pm 2.60}$ | $\mathbf{36.90 \pm 1.11}$ | $\mathbf{36.55 \pm 1.22}$ | $\mathbf{87.73 \pm 0.74}$ | $\mathbf{76.39 \pm 1.32}$ | $\mathbf{87.59 \pm 0.38}$ |
| $H_2$GCN | $74.05 \pm 4.22$ | $87.84 \pm 5.30$ | $85.69 \pm 3.51$ | $58.90 \pm 2.08$ | $28.44 \pm 7.14$ | $33.54 \pm 1.11$ | $88.15 \pm 1.22$ | $66.82 \pm 3.54$ | $\mathbf{89.44 \pm 0.48}$ |
| $H_2$GCN+GDC | $74.59 \pm 5.95$ | $87.03 \pm 5.24$ | $83.92 \pm 1.71$ | $60.79 \pm 1.27$ | $26.76 \pm 2.87$ | $33.43 \pm 1.26$ | $\mathbf{88.19 \pm 1.09}$ | $75.57 \pm 1.45$ | $89.42 \pm 0.55$ |
| $H_2$GCN+SDRF | $74.05 \pm 3.24$ | $71.35 \pm 4.71$ | $84.11 \pm 3.56$ | $58.60 \pm 1.78$ | $36.08 \pm 1.24$ | $32.83 \pm 0.78$ | $88.11 \pm 1.49$ | $\mathbf{76.85 \pm 1.47}$ | $\mathbf{89.44 \pm 0.51}$ |
| $H_2$GCN+QDC | $77.84 \pm 2.91$ | $87.46 \pm 4.17$ | $85.10 \pm 4.13$ | $\mathbf{61.52 \pm 1.91}$ | $36.30 \pm 0.96$ | $34.35 \pm 0.59$ | $74.76 \pm 2.29$ | $75.63 \pm 1.53$ | $73.34 \pm 3.10$ |
| $H_2$GCN+MultiScaleQDC | $\mathbf{76.01 \pm 3.72}$ | $\mathbf{88.38 \pm 4.84}$ | $\mathbf{86.86 \pm 3.51}$ | $59.14 \pm 1.61$ | $\mathbf{36.64 \pm 1.88}$ | $\mathbf{34.70 \pm 0.94}$ | $87.87 \pm 1.43$ | $76.13 \pm 1.79$ | $89.00 \pm 0.46$ |

CORA, CITESEER, and PUBMED. Where applicable, we use the same data splits as Pei et al. (2020). Results are then averaged over all splits, and the average and standard deviation are reported. These datasets represent a mix of standard heterophilic and homophilic graph datasets. The statistics for the datasets are presented in the first three rows of Table 1, where we have used the definition proposed by Pei et al. (2020) for homophily, given by:

$$\mathcal{H}(\mathcal{G}) = \frac{1}{|\mathcal{V}|} \sum_{v \in \mathcal{V}} \sum_{u \in \mathcal{N}_v} \frac{\mathbf{1}_{l(v)=l(u)}}{|\mathcal{N}_v|} \tag{8}$$

where $\mathcal{N}$ is the neighborhood operator and $l$ is the operator that returns the label of the vertex.

**Node Classification** We present the results from our experiments in Table 1. We observe that QDC provides improvements in accuracy across the heterophilic datasets, but seems to provide mixed results for Cora, Citeseer, and Pubmed. By using MultiScaleQDC, we see that multi-scale modeling appears to provide improvements across all datasets. This validates our hypothesis that QDC can provide a viable step forward to improving GNN performance. These results are consistent for both QDC and MultiScaleQDC modifications to our three base models – GCN, GAT, and $H_2$GCN.

**Analysis of Hyper-parameters** In addition to the hyperparameters associated with the underlying model (*e.g.* number of layers or number of hidden units), QDC has multiple hyperparameters, which are unique to the rewiring process that we tuned as part of our experiments. These hyperparameters correspond to $\mu$, the mean of the gaussian, $\sigma$, the standard deviation of the gaussian; and $k$, our cutoff parameter. This is not dissimilar from methods like GDC or SDRF. GDC includes hyperparameter for $\alpha$ and $k$ which correspond to the diffusion strength and the cutoff parameter, respectively. SDRF has hyperparameters that correspond to the maximum number of iterations; the temperature, $\tau$; and Ricci-curvature upper-bound, $C^+$. QDC only introduces one additional hyperparameter when compared with GDC, and has the same number of hyperparameters as SDRF. To understand the sensitivity of our method to these hyperparameters, we first present a violin plot in Figure 3, which plots a kernel density estimate of the model performances from the experiments on a GCN, GCN+QDC, and MultiScaleQDC. In the case of the Cornell dataset, we clearly observe that MultiScaleQDC has two humps, which correspond to the GCN and QDC distributions. We see similar patterns in the Texas, Wisconsin, Squirrel, and Actor datasets as well. This robust behaviour also holds for GAT based models, as can be seen from Figures 6. Furthermore, we clearly see that there are many experimental settings that out-perform the baseline model. While there are many effective experimental settings, we highlight that our method is still quite sensitive to choice of hyperparameter. Furthermore, each step of hyperparameter tuning is more expensive than in the case of a GCN or GAT because we have to compute new sets of approximate eigenvectors.

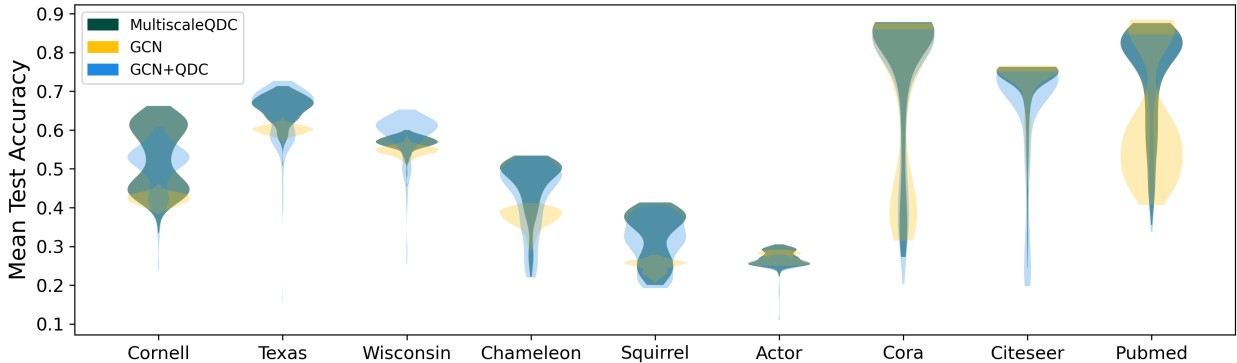

Figure 3: Violin plots of our experiments GCN+MultiScaleQDC (green), GCN (yellow), and GCN+QDC (blue), where these plots are generated by aggregating over all experiments associated with each model. We observe that both GCN+QDC and GCN+MultiScaleQDC generally have a high density of near-optimal configurations.

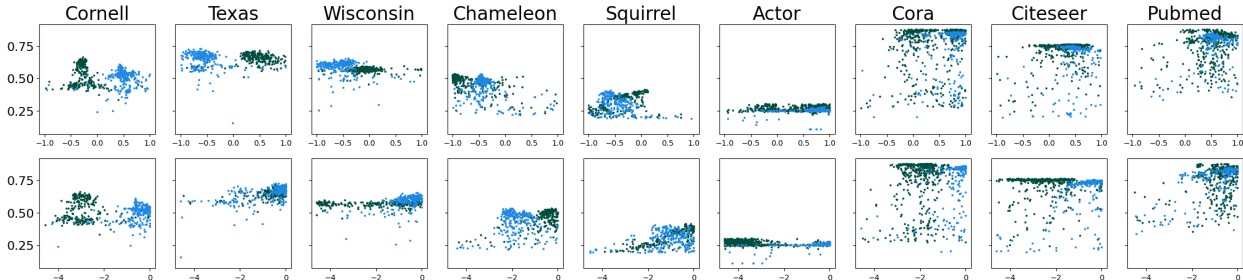

Figure 4: Scatter plots of mean test accuracy plotted against hyperparameters $\mu$ and $\ln(\sigma)$ in the first and second rows respectively for GCN+QDC (blue) and GCN+MultiScaleQDC (green). We observe that each of QDC and MultiScaleQDC are robust with respect to deviations in each of the hyperparameters.

We next turn our attention to the sensitivity of our model to $\mu$ and $\sigma$ for both QDC and MultiScaleQDC models by plotting mean test accuracy against $\mu$ and $\sigma$ in the first and second rows of Figure 4 respectively. We have plotted both GCN+QDC (blue) and our MultiScaleQDC (green) on the same plot. We observe that in general, there are many settings of $\mu$ and $\sigma$ that provide near equivalent performance which indicates that our method is robust to potentially suboptimal choice of hyperparameters. Interestingly, we find that the optimal $\mu$s for GCN+QDC and our MultiScaleQDC model are quite different. This is because in the MultiScaleQDC case, we are looking for eigenvectors that correct for any deficiencies in the original combinatorial Laplacian. In Figure 8 we present a 3D triangulated surface generated from the same data used to generate the scatter plots in Figure 4, so that we could better understand the correlations between both sets of hyperparameters. In this figure we find relatively wide plateaus of high performance that are consistent with the findings in Figure 4, although these surface plots are somewhat difficult to interpret without the aid of the 2d projections presented in Figure 4. We observe similar robust behaviour for GAT-based models as well, as can be seen from Figures 6, Figure 7, and Figure 9.

**Importance of Gaussian Filter** At the core of QDC is the choice of filter. In our development of our method we have chosen to use a Gaussian filter because it models inhomogeneous broadening, which is a physical effect that is observed as a transition frequencies taking on a Gaussian profile due to microscopic details of the system such as atomic motion. This physical model is intuitively sensible if we imagine that our vertices are analogous to the atoms, and the latent embeddings are the atomic positions. While we can provide physical arguments from analogy for this choice, citation networks are not molecular systems. This

Figure 5: Plots of the homophily as a function of the eigenvalues. We observe that homophily has a strong spectral dependence, and that the mid-band peaks in homophily agree with recovered optimal $\mu$s.

raises the question of whether the Gaussian form of our filter is important, or whether any band-pass filter would be sufficient. To answer this question we implemented a variant of QDC given by

$$\mathcal{B}(x_i, x_j) = \sum_\alpha \sigma\left(E_\alpha - \mu + \gamma\right) \sigma\left(\mu + \gamma - E_\alpha\right) \phi_\alpha^\dagger(x_i) \phi_\alpha(x_j), \qquad (9)$$

where $\sigma(\cdot)$ is the logistic sigmoid function, $\mu$ is the center of our bandpass filter, $\gamma$ is the width of our band-pass filter, and $\mathcal{B}$ is the band-pass version of QDC which we term the Band Pass Diffusion Convolution(BPDC). Using this filter, we performed experiments on a range of data sets using BPDC as our transition matrix with a GCN and have presented those results below in Table 1. We observe that BPDC is able to provide significant lift across the heterophilic datasets, but that lift is in general smaller than that observed with QDC.

**Spectral Dependence of Homophily** It has previously been observed that the performance of Graph Convolution models correlates with the homophily of the graph, which motivates us to ask whether homophily is spectrally dependent. To answer this question, we constructed adjacency matricies from subsets of the eigenvectors that corresponded to each unique eigenvalue. In the case where the eigenvalues were degenerate, we computed the mean homophily. We then sparsified the resulting adjacency matrix by removing all entries smaller than $1e-7$, and plotted the results in Figure 5. We observe that the homophily is highly spectrally dependent. `Actor` appears to be an outlier in this regard, with the optimal $\mu$ near 1 and the homophilic peaks existing in the range $[-1, 0.5]$. We attribute this to the generally poor performance on the Actor dataset, with a wide but flat performance envelope. In the case of the `Cornell` dataset, we observe that the dataset is generally quite heterophilic but becomes more homophilic in higher portions of the spectrum; and observe that the $\mu$ cluster for GCN+QDC in Figure 4 corresponds to this highly homophilic region. Similar trends are found for both `Texas` and `Wisconsin`. We observe spectral variations of homophily for `Chameleon` and `Cora` as well; and note the same agreement between the optimal $\mu$ and this observed spectral peaks in the homophily curves.

## 6 Conclusion

In this work we have introduced a quantum diffusion kernel that we have termed QDC, and a multiscale model that we have termed MultiScaleQDC. We have motivated this convolution kernel through a deep connection with quantum dynamics on graphs. In experiments we have shown that QDC generally helps in cases of heterophilic node classification, and MultiScaleQDC seems to improve in both homophilic and heterophilic node classifications settings when included in three different base GNN models – GCN, GAT, and $H_2$GCN. Additionally, we find that the improvements are robust across configurations and that Quantum inspired filters provide better results than more traditional square band-limited filters. We have additionally explored the spectral dependence of homophily, and found that not only is homophily spectrally dependent, but that this can explain the efficacy of our band pass filters.

**Limitations** While we are able to use iterative, matrix-free, eigensolvers, our method is still more expensive than spatial convolutions and limits the applicability of our method to large graphs. Additionally, propagating gradients through approximate eigensolvers is quite challenging, making it difficult to optimize the

parameters of the diffusion kernel during training time. Finally, because our method is spectral, we are only able to use this method in transductive settings. We believe that quantum convolution in the spatial domain will open up avenues to address these issues, and are excited to explore this approach in followup work.

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

# A   Appendix

## A.1   GAT Hyperparameter plots

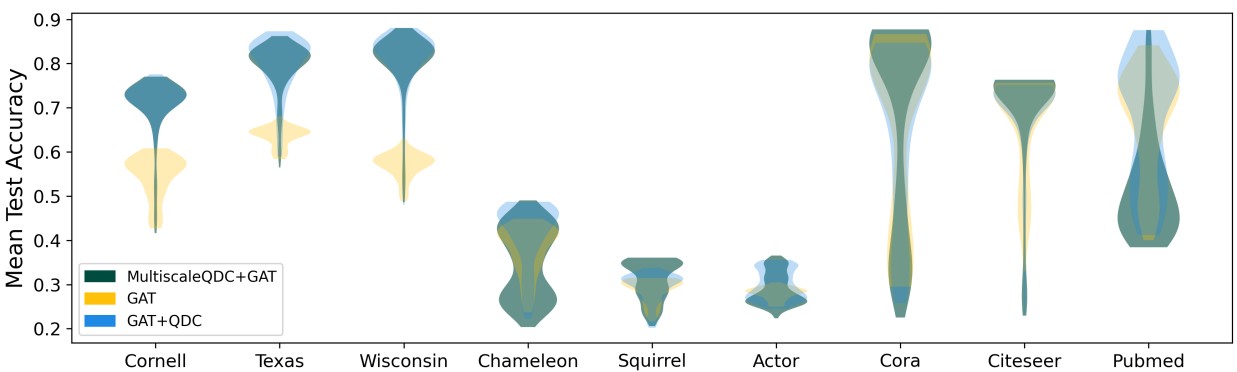

Figure 6: Violin plots of our experiments GAT+MultiScaleQDC (green), GAT (yellow), and GAT+QDC (blue), where these plots are generated by aggregating over all experiments associated with each model. We observe that both GAT+QDC and GAT+MultiScaleQDC generally have a high density of near-optimal configurations.

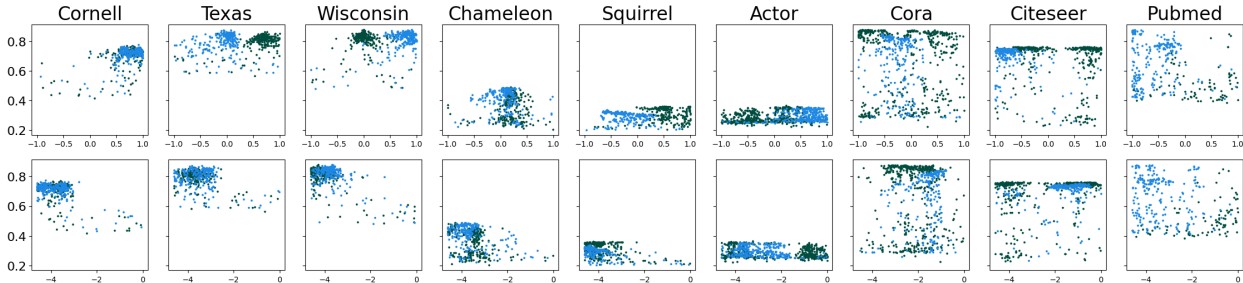

Figure 7: Scatter plots of mean test accuracy plotted against hyperparameters $\mu$ and $\ln(\sigma)$ in the first and second rows respectively for GAT+QDC (blue) and GAT+MultiScaleQDC (green). We observe that each of QDC and MultiScaleQDC are robust with respect to deviations in each of the hyperparameters.

## A.2   Runtime Costs

## A.3   Hyperparameter 3d Plots

## A.4   Model Details

We performed 250 steps of hyper-parameter optimization for each of the models presented in Table 1. All training runs were run with a maximum of 1000 steps for each split, with early stopping turned on after 50 steps. In the interest of reproducibility, we outline the parameters and ranges that we're optimized for each model below.

Table 2: Runtime costs for the QDC and MultiScaleQDC compared to a variety of baselines. We find that QDC requires a nontrivial increase in compute costs, but that this expense often carries with it a significant accuracy gain.

|  | GCN | GCN+GDC | GCN+SDRF | GCN + QDC | GCN + MultiScaleQDC |
|---|---|---|---|---|---|
| Cornell | 5.25 | 5.28 | 5.18 | 19.41 | 40.87 |
| Texas | 5.41 | 5.27 | 6.55 | 8.46 | 4.52 |
| Wisconsin | 8.47 | 8.98 | 12.02 | 31.88 | 22.56 |
| Chameleon | 7.42 | 39.45 | 52.92 | 20.22 | 257.33 |
| Squirrel | 36.65 | 32.42 | 160.73 | 77.12 | 115.47 |
| Actor | 12.98 | 10.80 | 332.40 | 309.55 | 299.91 |
| Cora | 20.08 | 83.71 | 42.13 | 79.52 | 166.70 |
| Citeseer | 36.90 | 59.25 | 51.02 | 59.25 | 130.39 |
| Pubmed | 81.92 | 84.62 | 4230.08 | 2366.10 | 2377.68 |

Table 3: Hyper-parameter ranges that we optimized over for our GCN.

| Parameters | Distribution | Values |
|---|---|---|
| Number of Layers | Categorical | [1, 2] |
| Hidden Dim Size | Categorical | [2, 4, 8, 16, 32, 64, 128] |
| Dropout percentage | Uniform | [0, 0.99] |
| Learning Rate | Loguniform | [1e-4, 1e-1] |
| Weight Decay | uniform | [0.0, 0.9] |

Table 4: Hyper-parameter ranges that we optimized over for our GCN+GDC.

| Parameters | Distribution | Values |
|---|---|---|
| Number of Layers | Categorical | [1, 2] |
| Hidden Dim Size | Categorical | [2, 4, 8, 16, 32, 64, 128] |
| Dropout percentage | Uniform | [0, 0.99] |
| GDC-$\alpha$ | uniform | [0.001, 0.5] |
| GDC-$\epsilon$ | uniform | [1e-7, 1e-1] |
| Learning Rate | Loguniform | [1e-4, 1e-1] |
| Weight Decay | uniform | [0.0, 0.9] |

Table 5: Hyper-parameter ranges that we optimized over for our GCN+QDC.

| Parameters | Distribution | Values |
|---|---|---|
| Number of Layers | Categorical | [1, 2] |
| Hidden Dim Size | Categorical | [2, 4, 8, 16, 32, 64, 128] |
| Dropout percentage | Uniform | [0, 0.99] |
| QDC-$\mu$ | uniform | [-1, 1] |
| QDC-$\sigma$ | uniform | [0.1, 1.0] |
| QDC-$\epsilon$ | loguniform | [1e-7, 1e-1] |
| Learning Rate | Loguniform | [1e-4, 1e-1] |
| Weight Decay | uniform | [0.0, 0.9] |

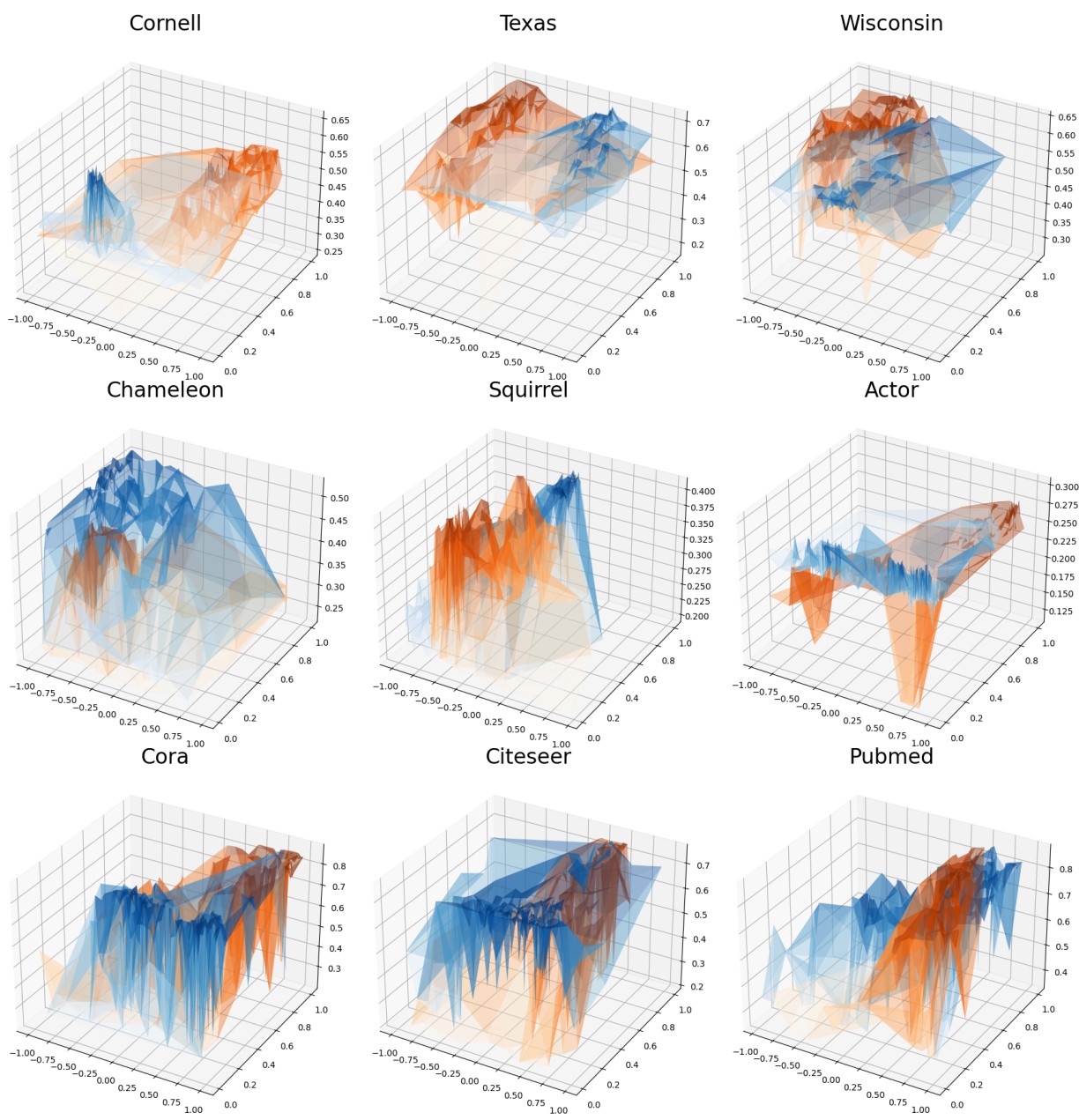

Figure 8: Triangulation surface plots of mean test accuracy plotted against hyperparameters $\mu$ and $\ln(\sigma)$ for GCN+QDC (orange) and GCN+MultiScaleQDC (blue). We observe that each of QDC and MultiScaleQDC are robust with respect to deviations in each of the hyperparameters because the high performance regions tend to be quite large.

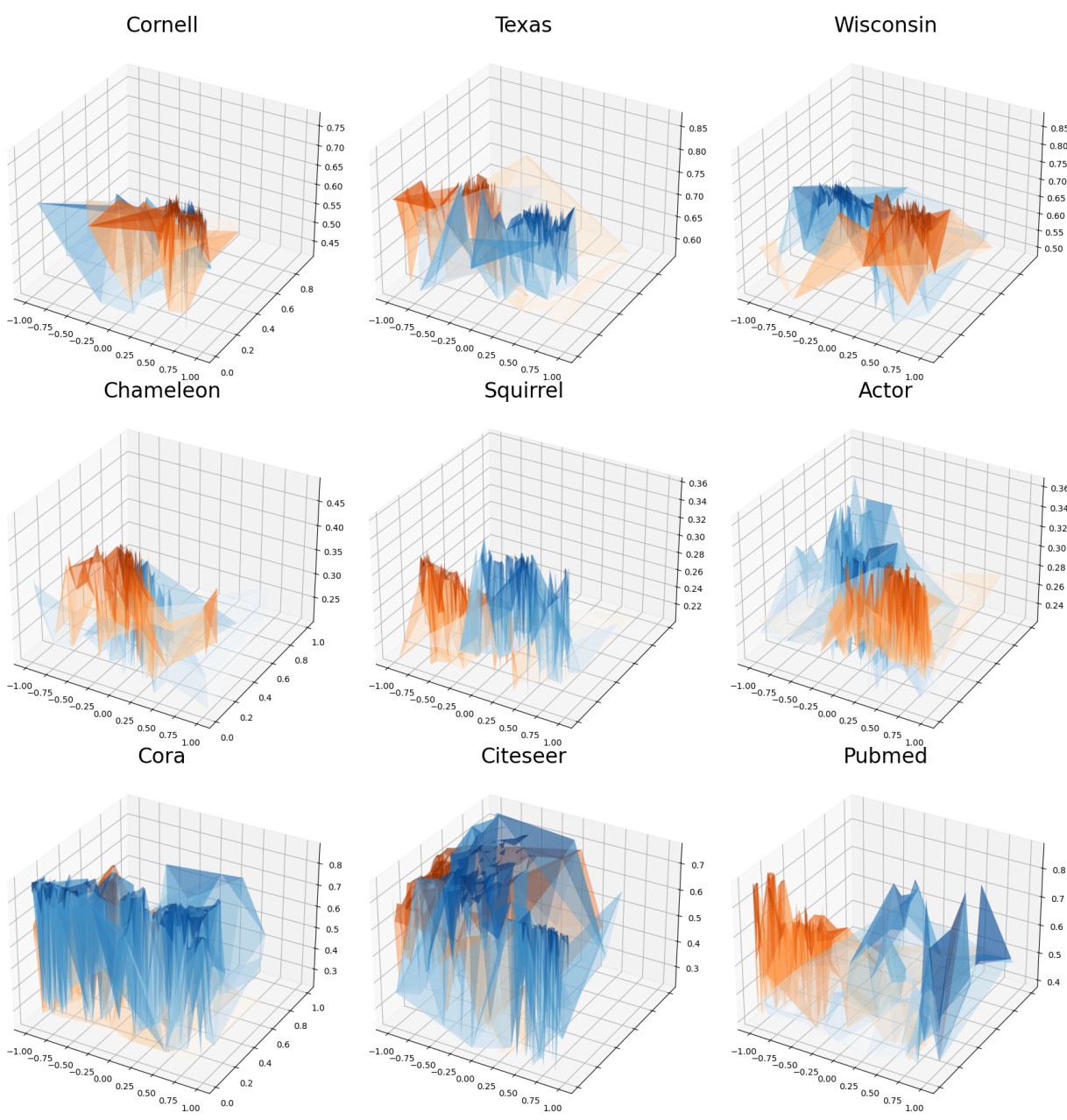

Figure 9: Triangulation surface plots of mean test accuracy plotted against hyperparameters $\mu$ and $\ln(\sigma)$ for GAT+QDC (orange) and GAT+MultiScaleQDC (blue). We observe that each of QDC and MultiScaleQDC are robust with respect to deviations in each of the hyperparameters because the high performance regions tend to be quite large.

Table 6: Hyper-parameter ranges that we optimized over for our MultiScaleQDC.

| Parameters | Distribution | Values |
| --- | --- | --- |
| GCN Number of Layers | Categorical | [1, 2] |
| GCN Hidden Dim Size | Categorical | [2, 4, 8, 16, 32, 64, 128] |
| GCN Dropout percentage | Uniform | [0, 0.99] |
| QDC Number of Layers | Categorical | [1, 2] |
| QDC Hidden Dim Size | Categorical | [2, 4, 8, 16, 32, 64, 128] |
| QDC Dropout percentage | Uniform | [0, 0.99] |
| QDC-$\mu$ | Uniform | [-1, 1] |
| QDC-$\sigma$ | Uniform | [0.1, 1.0] |
| QDC-$\epsilon$ | loguniform | [1e-7, 1e-1] |
| Combinator | Categorical | [ concat, add] |
| Learning Rate | Loguniform | [1e-4, 1e-1] |
| Weight Decay | Uniform | [0.0, 0.9] |

Table 7: Hyper-parameter ranges that we optimized over for our GAT.

| Parameters | Distribution | Values |
| --- | --- | --- |
| Number of Layers | Categorical | [1, 2] |
| Hidden Dim Size | Categorical | [2, 4, 8, 16, 32, 64, 128] |
| Number of Heads | Categorical | [1, 2, 3, 4, 5] |
| Dropout percentage | Uniform | [0, 0.99] |
| Learning Rate | Loguniform | [1e-4, 1e-1] |
| Weight Decay | Uniform | [0.0, 0.9] |

Table 8: Hyper-parameter ranges that we optimized over for our GAT+GDC.

| Parameters | Distribution | Values |
| --- | --- | --- |
| Number of Layers | Categorical | [1, 2] |
| Hidden Dim Size | Categorical | [2, 4, 8, 16, 32, 64, 128] |
| Number of Heads | Categorical | [1, 2, 3, 4, 5] |
| Dropout percentage | Uniform | [0, 0.99] |
| GDC-$\alpha$ | Uniform | [0.001, 0.5] |
| GDC-$\epsilon$ | Uniform | [1e-7, 1e-1] |
| Learning Rate | Loguniform | [1e-4, 1e-1] |
| Weight Decay | Uniform | [0.0, 0.9] |

Table 9: Hyper-parameter ranges that we optimized over for our GAT+QDC.

| Parameters | Distribution | Values |
| --- | --- | --- |
| Number of Layers | Categorical | [1, 2] |
| Hidden Dim Size | Categorical | [2, 4, 8, 16, 32, 64, 128] |
| Number of Heads | Categorical | [1, 2, 3, 4, 5] |
| Dropout percentage | Uniform | [0, 0.99] |
| QDC-$\mu$ | Uniform | [-1, 1] |
| QDC-$\sigma$ | Uniform | [0.1, 1.0] |
| QDC-$\epsilon$ | loguniform | [1e-7, 1e-1] |
| Learning Rate | Loguniform | [1e-4, 1e-1] |
| Weight Decay | Uniform | [0.0, 0.9] |

Table 10: Hyper-parameter ranges that we optimized over for our Multiscale GAT+QDC.

| Parameters | Distribution | Values |
|---|---|---|
| GAT Number of Layers | Categorical | [1, 2] |
| GAT Hidden Dim Size | Categorical | [2, 4, 8, 16, 32, 64, 128] |
| GAT Number of Heads | Categorical | [1, 2, 3, 4, 5] |
| GAT Dropout percentage | Uniform | [0, 0.99] |
| QDC Number of Layers | Categorical | [1, 2] |
| QDC Hidden Dim Size | Categorical | [2, 4, 8, 16, 32, 64, 128] |
| QDC Number of Heads | Categorical | [1, 2, 3, 4, 5] |
| QDC Dropout percentage | Uniform | [0, 0.99] |
| QDC-$\mu$ | Uniform | [-1, 1] |
| QDC-$\sigma$ | Uniform | [0.1, 1.0] |
| QDC-$\epsilon$ | loguniform | [1e-7, 1e-1] |
| Combinator | Categorical | [ concat, add] |
| Learning Rate | Loguniform | [1e-4, 1e-1] |
| Weight Decay | Uniform | [0.0, 0.9] |

