# OpenReview forum: "QDC: Quantum Diffusion Convolution Kernels on Graphs"
_TMLR — Accepted by TMLR_

### Review · Reviewer_4R5C · 2023-08-17

**Summary Of Contributions:**

Gasteiger et al. 2019 presented an innovative method named "graph diffusion convolution" that primarily restructures a graph through generalized graph diffusion combined with sparsification and subsequent message passing on the altered graph (rewired graph). Building on this foundational approach, the current research introduces a new "quantum diffusion" technique for the graph rewiring process. This adaptation employs a Quantum diffusion kernel for graph diffusion, unlike Gasteiger et al.'s heat kernel-based method. This modification results in the formation of band-pass filters (Gaussian filters), as opposed to the low-pass filters (polynomial filters) created by the heat kernel.  Importantly, the study reveals that this quantum diffusion approach can synergize with existing low-pass filters, like the Laplacian, to conceive a multiscale method. Through empirical evaluation, the paper underscores the superiority of their proposed method compared to other graph diffusion convolution techniques.

**Audience:**

Yes

**Claims And Evidence:**

No

**Requested Changes:**

- **Writing Refinements**:

    - The introduction could be supported with a more compelling rationale.
    - A more comprehensive background on quantum diffusion and the heat equation is warranted in the introduction.
    - Positioning the method within the framework of Gasteiger et al. (2019) might not only simplify the presentation but also clearly delineate this research's contributions.
    - A concluding segment in the experiments section, detailing main takeaways and limitations would be advantageous.
- **Extended Comparisons**: Incorporating contemporary techniques that address heterophily in graphs would certainly underline this research's real-world relevance.

- **Addressing Issues**: The authors should explore potential solutions to the challenges posed by the additional hyperparameters and scalability concerns.

- **Additional Analysis**: A deeper investigation into the disparities between BPDC's and QDC's efficiencies is recommended. Understanding the implications of spectral dependence in homophily, especially in terms of model generalizability or potential biases, would also add depth to the analysis.

**Strengths And Weaknesses:**

### Strengths
- The research showcases that the proposed technique surpasses several other graph diffusion convolution methods, especially in tasks with heterophily.
- The insights on the spectral independence of homophily are interesting, albeit not exhaustive.

### Weaknesses
- **Conceptual Complexity**: The authors approach their method through the quantum dynamics prism for message passing kernels. While this viewpoint offers novelty, it might add layers of intricacy that could pose challenges in both understanding and implementation, especially for those lacking a quantum mechanics foundation. Simplifying the presentation of this quantum diffusion, which essentially utilizes Gaussian filters instead of the polynomial filters as seen in Gasteiger et al. (2019), could enhance clarity.

- **Presentation and Writing**: The introductory query, "Can we improve message passing in graph neural networks with a quantum mechanical message passing kernel?", lacks persuasive force and doesn't compellingly justify the need for quantum diffusion. Additionally, the introduction's explication of quantum diffusion and its nuances with the heat equation appears somewhat opaque. This may pose comprehension challenges for readers not well-versed with the topic. Moreover, the experiments section would benefit from a summarizing segment highlighting primary outcomes, limitations, and prospective avenues.

- **Baseline Comparisons**: The empirical evaluations indicate that the band filters from the proposed quantum diffusion can notably improve GNNs, particularly in contexts of heterophily or low homophily. Given that this theme has been the focus of recent works, including [H2GCN](https://arxiv.org/abs/2006.11468) and [Bodnar et al. 2022](https://proceedings.neurips.cc/paper_files/paper/2022/file/75c45fca2aa416ada062b26cc4fb7641-Paper-Conference.pdf), a broader comparative analysis would enhance the discernment of this method's practical value.

- **Hyperparameters Concern**: In comparison to the heat kernel, the introduced quantum diffusion kernel necessitates two extra hyperparameters, $\mu$ and $\sigma$. It is clear from Figure 4 that the model's efficacy is considerably influenced by these hyperparameters, which implies the need for meticulous hyperparameter optimization.

- **Scalability Issues**: Calculating the quantum diffusion kernel, as proposed, seems computationally demanding for large graphs.

- **Analytical Gaps**: A more in-depth discourse or exploration into why BPDC's efficiency is predominantly less than QDC's would be valuable. Further, understanding the implications of the spectral dependence of homophily on model generalizability or potential biases would be insightful.

---

> ### Author Response · Authors · 2023-08-18
> **Thank you for your review, Quick Response**
>
> Thank you for your review of our work, and your highly detailed feedback. We would like to incorporate your feedback into our manuscript, and to that end we are planning to do the following to address your points. **Please let us know if the proposed work is not sufficient.**
>
> **Conceptual Clarity:**
> - We will rework the introduction to provide a more compelling rationale for this study. In particular, we will provide a brief comparison between the two types of dynamics. Additionally, we will cut the discussion of Dirac notation and some of the components of quantum dynamics, so that we can focus more on comparing and contrasting quantum and heat dynamics. We will make the discussion oversmoothing and bottlenecks dynamics more clear as well.
> - A question for the reviewer -- Do you feel as if figure 2 was clear and added to the discussion?
>
> **Presentation and Writing:**
> - By introducing heat diffusion in the intro, we will rework the introductory query to be more persuasive.
> - We will summarize our experimental results more clearly both in the conclusion section as well as in the experiments section.
>
> **Hyperparameters Concern:**
> - We will explicitly mention the hyperparameters introduced both by GDC and SDRL, and provide similar hyperparameter analysis in the appendix.
> - While we do agree that the introduction of hyperparameters introduces complexity, we disagree that extensive and meticulous hyperparameter tuning is necessary. In figure 4 we observe quite wide performance bands for both $\mu$ and $\sigma$ indicating that getting the hyperparameters directionally correct is usually enough. We will make this point make this point more clear in the text.
> - Thank you for raising these points. We agree that addressing them would make the paper more clear. **Would doing so satisfy your concerns?**
>
> **Scalability issues:**
> - We that even though we make use of a matrix-free eigensolver, our method is still more expensive than a spatial convolution. We have highlighted this in the conclusion, but will rework the efficient diagonalization heading to make more clear that matrix-free eigensolvers are still expensive. In addition, we will include a runtime table that splits out the pre-processing and training time for all methods in the appendix. **Would you find this satisfactory to address your scalability concerns?**
>
> **Analytical Gaps:**
> - We will compute the spectral dependence of homophily for all datasets under consideration and include those plots in the appendix if they will not fit in the main text.
> - We will explore the spectral differences between optimal BPDC and QDC filters on multiple datasets to elucidate their differences in efficacy.
>
> **Baseline Comparisons:**
> - We opted to compare QDC to SDRL and GDC because these methods each involved preprocessing and rewiring of the Adjacency matrix, and could be used with many different models such as GCN, GAT, a Graph Transformer, or H2GCN. We will make our rationale for our choice of baseline more clear in the text.
> - We do agree that our discussion of heterophilic graphs could warrant additional model comparisons, although we intended to focus on the differences in performance of the rewiring techniques rather than the models themselves. We will run additional experiments with H2GCN with and without QDC and MultiscaleQDC.

---

> > ### Comment · Reviewer_4R5C · 2023-08-21
> > **Response and clarification**
> >
> > Thank you for you quick response.
> > >On the clarity of Figure 2
> >
> > - Figure 2 lacks clarity in its presentation. Although there are four distinct colors depicted, there is no indication of what the orange and purple hues signify.
> >
> > >Hyperparameters concern
> >
> > - From figure 4, it's evident that performance exhibits significant variability even in proximity to the optimal values of $\mu$ and $\ln(\sigma)$. Taking the Cornell dataset as an instance, the optimal $\mu$ value for MultiscaleQDC hoevers around -0.3. Yet, values near this optimum display a pronounced variation: some yield an accuracy of just 0.5 while the optimal $\mu$ choice achieves a 0.66 accuracy.
> > - I'm perplexed by the decision to employ scatter plots. Wouldn't it have been more informative to present a curve with error bars?
> > - Another point of contention lies in performance portrayal. Given that the best accuracies were achieved using GAT+QDC/MultiScaleQDC, why does this figure exclusively depict accuracies for GCN+QDC/MultiScaleQDC?
> >
> > >Scalability issues
> >
> > - A runtime comparative analysis from the authors would be invaluable. Moreover, integrating experiments on the [ogb node property prediction tasks](https://ogb.stanford.edu/docs/nodeprop/) would certainly enhance the paper's practical impact.

---

> > > ### Author Response · Authors · 2023-08-21
> > >
> > > Thank you for taking the time to clarify your point and for being a highly engaged reviewer. We feel much better equipped to address these points in full, thereby making the paper stronger in the process.
> > >
> > > **Figure 2:** We will add a colour bar, which should make the figure a little easier to interpret
> > >
> > > **Hyperparameters:** I understand your concern now. Many of those points correspond to tuning different configurations of the model (e.g. number of GCN layers, size of hidden dimension, dropout percentage, learning rate, etc) and not QDC itself. We will explore alternative plotting strategies to make this clear. There was also no reason to choose GCN over GAT, and we will include both sets of figures, moving one to the appendix for space.

---

### Review · Reviewer_bvxn · 2023-09-04

**Summary Of Contributions:**

This paper proposes a new notion of convolution on graphs. The classical definition of convolution on graphs is done thanks to the (combinatorial) Laplacian, where the eigenvectors of the Laplacian define the Fourier modes. As a result, we can define filters on graphs as polynomials of the Laplacian. Here, the authors propose to replace the Laplacian with a Quantum Diffusion Kernel.

**Audience:**

Yes

**Claims And Evidence:**

No

**Requested Changes:**

As explained above, the paper requires major changes before any possible publication.
Other minor changes: in the definitions of matrices on page 3, use $\times$ instead of $x$.

**Strengths And Weaknesses:**

Section 3 about the background and the classical definition of convolutions for graphs is well-written and should probably be shortened. But Section 4 is completely unclear to me and the authors need to provide more details about their methodology. The authors introduce concepts from physics without explaining them, like "ket". I think that these concepts are not necessary. As graphs are finite, we are always dealing with finite dimensional states or operators, hence, it should be able to rephrase these concepts in terms of matrices and vectors.
As I understand it, the notion of Quantum Diffusion Convolution Kernels is a small modification from previous work. It looks like the "ket" $\phi$ in equations (8), (9) and (10) are the eigenvectors of $\mathcal{H}=-\Delta$ the combinatorial Laplacians. However, I am unsure how to interpret the $x_i$ in the matrix $\mathcal{Q}(x_i,x_j)$.
I do not understand at all how this QDC can be encoded as a message-passing algorithm. This is perhaps explained in the paragraph Sparsification on page 7, but this is very unclear.
In the experiments, the authors claim that QDC has multiple hyperparameters, but they are not defined. To me, there are only 2 parameters $\mu$ and $\sigma$ appearing in equation (10) which are tuned thanks to SGD like the parameters $\theta_k$ in equation (5).

---

> ### Comment · Action_Editors · 2023-11-22
> **submit recommendation?**
>
> Dear reviewer bvxn,
>
> Could you please submit your final recommendation for this paper based on the authors response?
>
> Thank you, AE

---

### Review · Reviewer_MqtD · 2023-10-18

**Summary Of Contributions:**

The aims of this paper of this paper are to devise a new diffusion kernel to be used in graph neural networks. Graph diffusions were introduced as a means of overcoming expressivity issues in the spatial convolutions used in graph convolutional networks.

The main idea stems from considering a means of constructing a band pass filter on a graph. The authors motivate that common parametric realizations of filters, e.g. such as the Chebyshev expansion given in equation 5, can be seen as the generators governing some pde, connecting these methods to diffusion equations. This draws direct analogy to the heat equation, which, unfortunately, has an asymptotic time evolution which equilibrates unfavorably with the respect to the graph. They seek to highlight (and what they should express more explicitly) is that the choice of diffusion can have broad effect on the information propagation on the graph. There are many such diffusions that could be realized on the graph (and they mention some, but mainly focus their proposal in juxtaposition with the heat kernel which seems especially ill-suited for the task).

In lieu of the heat diffusion, they consider dynamics under the Schrodinger equation. They state this equation gives more favorable message-passing dynamics. Time evolution under the schrodinger equation is governed by a unitary operator which is the matrix exponential of the Hamiltonian of the system. They study the case for the hamiltonian of a free particle, which has eigenstates which are linear superpositions of basis vectors $\ket{\phi_i}$ weights by coefficients $c_i$ and a phase factor $e^{iE_it}$ (which, in computing likelihoods via the Born rule, should vanish only for terms corresponding to $i=j$ in the inner products $\bra{\phi_j}\ket{\phi_i}$).

Simulating the time evolution of quantum systems is an open challenge in the field of quantum physics (when scaling with respect to the number of particles in the many-body system and with respect to the extent of the time evolution, as in practice, most time-evolutions of hamiltonians need to be Trotterized, introducing finite error which compounds wrt to the time evolution T), so it is a bit interesting to see folks suggest it as a means of getting at graph neural networks.

The reason they can get away with this and the quantum community cannot is because they circumvent the need to do the time-evolution by considering the infinite time dynamics of the system, which is made possible by the fact that they are considering the hamiltonian of a free particle.

However, this means that there are non-zero transition probabilities for all vertices and the transition matrix is dense. Diagonalizing this transition matrix is a challenge (as someone who is less familiar with graph conv nets, how large is N in practice?), but they can get away with iterative methods that give them access to a subset of the eigenvalues and vectors needed for the "frequencies" (energy states) associated with their filter.

They benchmark their method using some standard datasets and compare it to GCNs. I am not an expert in GCNs and do not know if there are other methods they should reasonably be comparing to other than the ones they've listed. What they do compare to shows that their method performs quite well.


I detail what I see to be the strengths and weaknesses in the other sections, but, if they authors address my remarks made there, I think the paper should be **accepted.**

**Audience:**

Yes

**Broader Impact Concerns:**

None.

**Claims And Evidence:**

Yes

**Requested Changes:**

- Address the typos described in expository weaknesses
- Consider removing the stress on this being a "quantum" method -- indeed there is very little quantum about it (many-body entanglement, meaningful potentials, etc) other than using the PDE associated with quantum mechanics and deriving things under unitary evolution (this problem is entirely classical, for all intents and purposes).
- State more clearly that what Hamiltonian you are actually using -- you discuss the free particle but because of my own failings or because of lack of specification, I was a bit lost when you jumped to experiments (but assumed you stuck with the free particle b/c you needed to get to equation 10)
- State clearly how many basis vectors you are assuming to use in equation 8 :)

**Strengths And Weaknesses:**

Strengths:
- The exposition motivating both the reason for a diffusion perspective on the design of GCNs and the utility for the Schrodinger equation in this context is well-founded. The document is overall very readable in this sense, other than for the related works section (which I discuss in more detail below) and a few things in Section 3.
- The experimental results clearly highlight that the preconditioned GCN with the Schrodinger method is performant.


Expository Weaknesses:
- Numerous acronyms introduced without any clarification in the related works. e.g., the labelling +FA is undefined in relation to the paper of Alon and Yahav 2020 that is cited. Same goes for "SDRL", "GRAND" method, "BLEND", "PDE-GCN", etc.
- The name ChebNet is introduced in the paragraph below equation (5) without any definition.
- Caption of Figure 2 does not seemingly support the evidence of the figure. While it is true that the the top cluster on the bar-bell still has some mixing at time t= 1000, the bottom cluster is entirely uniform. I see what the authors intend to stress with information bottleneck of the head diffusion, but they should just clarify the text in the caption to be a bit more nuanced.
- Typo in first sentence of Section 4: Laplaician --> Laplacian.

Technical Weaknesses:
- It's important to point out that this paper does not actually model any sort of quantum dynamics. Indeed, this is a bit of a distraction from, say, just specifying that you write down the Schrodinger equation (just another PDE) and consider the infinite time limiting dynamics of the solution to the Schrodinger equation for a free particle. Really, the rest of this does nothing quantum. It's more distracting than elucidating to label it as a quantum diffusion -- there is no entanglement, seemingly no
- The authors point out (which is useful to state!) that a priori having a dense kernel is a blessing (because it helps overcome message passing bottlenecks) and a curse (because it makes all diagonalizations etc very costly). This isn't really a weakness in the paper, it's just being realistic, which is a plus in my book.

---

> ### Author Response · Authors · 2023-10-18
>
> Thank you for your thorough review of our work. We would like to incorporate your feedback into our manuscript, and to that end we are planning to do the following to address your points. **Please let us know if the proposed work is not sufficient.**
>
> - We will address the typos that were pointed out, and perform additional proof reading to find any additional ones.
> - We will combine the feedback from reviewer 4R5C with yours, and rework the theory section to simplify the discussion of the role of quantum dynamics. This will include explicitly stating our Hamiltonian. We believe that retaining references to the quantum nature is important because it motivates work that is in progress that explores both tuning of potentials and finite time propagation, but we will make more explicit that we are not leveraging many of the resources that typically make quantum methods more powerful.
> - We will state more clearly how many basis vectors we are using in equation 8.
> - We will refine the related work section to make it a bit less of an alphabet-soup of acronyms.

---

### Author Response · Authors · 2023-11-01
**Revisions**

We would like to thank all reviewers for your valuable feedback on this work. We have incorporated the feedback into our paper and feel that it is much stronger as a result. We summarize the changes below:

- We have fixed minor typos throughout the paper.
- We have explicitly mentioned the number of basis vectors used in Equation 8 as part of our discussion of efficient diagonalization
- We have added a table to the appendix that presents the observed average runtime for our experiments, and referenced this table in our discussion of efficient diagonalization. We have also taken care to note that while faster than traditional eigen-solvers, matrix free eigensolvers are still expensive.
- We have added a color bar to figure 2
- We have simplified the discussion of quantum mechanics in the work, and removed the definition of Dirac notation. We have additionally made more clear that $x_i$ is defined as the position of the $i^{th}$ vertex.
- In simplifying our presentation of our method, we have also explicitly stated the equation for the free particle on a graph in equation 6
- We have clarified our discussion of hyperparameter sensitivity. Specifically, we have explicitly stated our hyperparameters that are tuned in section 5, heading "Analysis of hyperparameters". We note that neither $\mu$ nor $\sigma$ can be readily tuned with SGD in this formulation of the work.
- We have reworked the introduction to better motivate our study, as well as to provide an introduction to both quantum and thermal dynamics.
- We explored different plotting strategies to better present the hyperparameter sensitivity plots but found nothing that was absolutely better. We have elected to generate large 3d-triangulation plots of the surface in the appendix, which should hopefully provide a clearer view of our sensitivity to hyperparameters. A discussion of these plots was added to the main text.
- We have included GAT versions of Figures 3 and 4 in the appendix, and remarks in the main text.
- We have computed the spectral dependence of homophily for all datasets and included those plots in the the main text. Update: the strategy worked. We have bypassed the computation limits
- We have added citations for H2GCN and the relevant github repos that we used in this study.
- We have included experiments for H2GCN, H2GCN + GDC, H2GCN + SDRF, H2GCN + QDC, and H2GCN + MultiScaleQDC in the main table, and expanded our discussion of these datasets.

We thank you all again for your thoughtful reviews. Sincerely,

The authors

---

### Decision · Action_Editor_L8bC · 2023-12-14

**Recommendation:** Accept with minor revision

**Comment:**

The paper studies GNNs, in particular graph convolutions based on graph diffusions, as studied by Gasteiger et al. 2019 and other works. In order overcome the over-smoothing limitations of heat diffusions that were considered in most prior work, they consider a different physical model inspired by quantum diffusion, leading to new diffusion-based graph convolutions that are more localized/band-pass.

The reviewers appreciated the method, its empirical effectiveness, and its motivation, though they generally found the quantum jargon to be a bit too extensive and in some cases unnecessary. After discussion, the author addressed this in large part, making the manuscript significantly stronger. As a result, the paper is now in much better shape, and I recommend acceptance.

In his final decision, reviewer 4R5C provided the following additional requests, which I ask the author to address in a final minor revision:
> * Clarity and Rationale in Writing: The authors have made attempts to enhance the manuscript's clarity in this revision. Nonetheless, further improvements are necessary, especially in the introduction. The question posed, "Can we improve graph neural networks by considering a different physical model?" still seems insufficiently justified. The core of this study appears to be the application of band-pass filters (Gaussian filters) instead of the low pass filters (polynomial filters) in the graph rewiring process. This distinction needs to be more prominently articulated, rather than motivating the propose method from a physical perspective.
> * Quantum Mechanics Integration: Both Reviewer MqtD and bvxn have noted, and I agree, that the introduction of quantum mechanics into the methodology seems to add unnecessary complexity. The approach largely builds upon Gasteiger et al.'s 2019 framework, and this connection seems to be sufficient to present the proposed method.
> * Experimental Section Structure: The experimental section currently jumps directly into the setup without adequately outlining the objectives or summarizing key findings.
> * Hyperparameter Optimization: Figure 4 indicates a significant impact of hyperparameters on the model's performance, suggesting a need for sophisticated optimization. The method also appears to use more exhaustive search grids (as per Tables 6-10) compared to baseline methods, potentially leading to unfair comparison.
> * Scalability and Computational Complexity: The manuscript should address concerns regarding scalability and computational complexity. The method's application to large real-world graphs seems computationally intractable, and the additional time required for hyperparameter tuning exacerbates this issue.

Also, please respond to reviewer bvxn.

**Audience:**

Yes

**Claims And Evidence:**

Yes